# Changes in the Serum Levels of Cytokines: IL-1β, IL-4, IL-8 and IL-10 in Depression with and without Posttraumatic Stress Disorder

**DOI:** 10.3390/brainsci12030387

**Published:** 2022-03-14

**Authors:** Ewa Ogłodek

**Affiliations:** 1Department of Health Sciences, Jan Dlugosz University in Czestochowa, Armii Krajowej 13/15 Street, 42-200 Czestochowa, Poland; e.oglodek@wp.pl; Tel.: +48-669-300-460; 2Department of Psychiatry, Collegium Medicum in Bydgoszcz, Nicolaus Copernicus University in Toruń, 87-100 Toruń, Poland

**Keywords:** cytokine, depression, interleukin, neuroprotection, posttraumatic stress disorder

## Abstract

Background: Both depressive disorders (DD) and post-traumatic stress disorders (PTSD) are caused by immune system dysfunction. Affected individuals show increased proinflammatory cytokine concentration levels. Also, it has been hypothesized that DD and PTSD might be associated with a generalized proinflammatory cytokine signature. The study assessed the concentration of IL-1β, IL-4, IL-8 and IL-10 in depression alone and with PTSD. Methods: The study involved 460 participants. Out of them, 420 subjects comprised a study group and 40 subjects comprised a control group. Each study group consisted of 60 patients with mild depression (MD), moderate depression (MOD), severe depression (SeD), MD and PTSD (MD + PTSD), MOD and PTSD (MOD + PTSD), SeD and PTSD (SeD + PTSD), and with PTSD alone. All patients had serum concentration of IL-1β, IL-4, IL-8 and IL-10 measured with ELISA. Results: DD and PTSD are reflected in IL-1β, IL-4, IL-8 and IL-10 concentration levels. It was reported that mean levels of IL-1β, IL-4, IL-8 increase as depression became more severe. A regular decrease in IL-10 concentration levels was noted with the onset and exacerbation of depressive symptoms. Conclusion: The findings might be useful when considering chronic inflammation as a potential target or biomarker in depression and PTSD treatment.

## 1. Introduction

Depression has been reported to be a global disease affecting people in all communities worldwide [1]. Moreover, it tends to co-occur with other disorders, the most common of which is chronic PTSD. Having experienced a traumatic event, the vast majority of individuals report symptoms such as intrusive thoughts or dreams about the event, flashbacks, irritability, hyper-alertness, as well as disrupted sleep, memory and/or concentration. The proper functioning of the immune system and the occurrence of the clinical symptoms of PTSD and/or depressive disorder (DD) are interdependent. Both inflammatory diseases and DD share a common risk factor, namely the activation of the immune system. Inflammatory changes in the body due to conditions other than PTSD or DD increase the risk of developing either PTSD or DD. On the other hand, the primary onset of DD and/or PTSD increases the risk of developing other inflammatory conditions, e.g., metabolic diseases. The role of the pro-inflammatory phenotype, which creates a potential genetic basis for the elevated risk of DD and/or PTSD, seems to be decisive in terms of the order of developing either disease. The pathophysiology of both DD and PTSD involves abnormalities in the HPA-axis. The chronic and repeated activation of this system leads to the detrimental effects of stress on the brain. Based on the glucocorticoid-hippocampal atrophy model, the glucocorticoids released during stress exert neurotoxic effects on the central nervous system (CNS). The hippocampus is an immensely susceptible organ due to the high density of glucocorticoid receptors. Since glucocorticoids regulate inflammatory responses, the HPA-axis is functionally linked to the immune system, and thus increased inflammation causes stress system activation. Indoleamine 2,3-dioxygenase (IDO), (Han et al., 2015) is claimed to be a factor that links “cytokine theory” with “monoamine theory” in DD [2,3].

The inflammatory markers of DD induce IDO, an enzyme involved in the process of tryptophan catabolism, to form compounds called TRYCATs. In turn, the IDO induction leads to a decrease in tryptophan and serotonin concentration levels, for which tryptophan is a precursor. TRYCATs, such as kynurenine and nitric oxide synthase (NOS), are produced as a result of tryptophan degradation. Kynurenine causes both anxiety and depression. Nitric oxide (NO), stimulated by NOS, may cause the excessive production of NO, which plays an important physiological role and is also a free radical that contributes to lipid peroxidation and arachidonic acid cascade activation. This leads to the increased production of other inflammatory markers, including prostaglandin E2, [4]. Another process observed in depression is associated with the ability of cytokines to increase glutaminergic transmission. One of the effects of increased glutaminergic transmission is reduced production of a brain-derived neurotrophic factor (BDNF). As a result, the increased glutaminergic transmission and reduced BDNF levels lead to changes in neuronal plasticity [5].

Cytokines have been found to cross the blood-brain barrier [6]. Therefore, they are involved in many aspects of depression pathophysiology, including neurotransmitter metabolism, neuroendocrine function, neurogenesis, neuronal integrity, synaptic remodelling, and neural plasticity.

Immune cytokines such as IL-1β, IL-4, IL-8 and IL-10 are reported to impact key processes related to DD development, including neuroplasticity, neurotransmission, oxidative stress, and neuro-endocrinological functions.

With regard to interleukin 8 (IL-8/CXCL8), it is said to show pleiotropic properties in terms of chemotaxis and pro-inflammatory effects that support both the activation and degranulation of neutrophils, basophils and monocytes/macrophages. The role of high circulating IL-8 levels is to decrease the infiltration of neutrophils to the inflammatory site. This allows IL-8 to take on either a pro- or anti-inflammatory role, depending on its concentration [7].

Proinflammatory interleukin 1-beta (IL-1β) has been shown to inhibit long term potentiation (LTP), a phenomenon associated with synaptic plasticity which is the biochemical basis of memory [8]. It is believed that this can be achieved by the ability of IL-1β to increase the production and accumulation of reactive oxygen species and JNK stress induced kinase (c-Jun N-terminal kinase), which can lead to both cell malfunction and apoptosis.

Interleukin (IL-4) exerts a number of biological effects, including the stimulation of activated B-cell and T-cell proliferation, and the differentiation of B cells into plasma cells. Additionally, its role is to regulate both humoral and adaptive immunity. Another function of IL-4 is to decrease the production of Th1 cells, macrophages, IFN-gamma, and dendritic cells IL-12. Certain studies of IL-4 in macrophages, found that it showed anti-inflammatory effects when administered either concurrently or shortly after an inflammatory stimulus [2]. Moreover, it was able to downregulate the production of inflammatory cytokines such as TNF. It is important to note that IL-4 is not purely an anti-inflammatory agent, but with proinflammatory stimulation, it can induce an enhanced inflammatory response [9].

Interleukin-10 (IL-10), also referred to as a T-helper lymphocyte type-2 cytokine, is a prime cytokine involved in the downregulation of inflammatory reactions. IL-10 is capable of suppressing the production of proinflammatory cytokines [10]. Additionally, it plays a part, among others, in the regulation of neurogenesis, neuroprotection or modulation of memory processes by influencing synaptic plasticity processes in the hippocampus [11]. Therefore, it can be assumed that IL-10 exerts neuroprotective effects and is involved in post-injury plasticity processes. The latest studies indicate that the administration of IL-10 can inhibit morphological changes associated with glia activation, the production of proinflammatory cytokines, and enzyme activity involved in the production of inflammatory mediators and free oxygen radicals.

In previous studies, the author of this publication determined other cytokines such as iNOS, HO-1, IL-33, MIP-1β, MIP-1 α, PMN Elastase, MDA, and IL-12, PON-1, GR, IL-18, and OxLDL in the same study groups [12,13,14]. The immune effects of IL-1β, IL-4, IL-10 and IL-8 in depression or PTSD have also been reported by other authors [15,16,17,18,19]. However, there are no studies that have evaluated cases with DD comorbid with PTSD. Thus, this study is a continuation of earlier research.

The aim of this study was to assess changes in the concentration of IL-1β, IL-4, IL-8 and IL-10 among patients with mild depression (MD), moderate depression (MOD), severe depression (SeD), MD and PTSD (MD + PTSD), MOD and PTSD (MOD + PTSD), SeD and PTSD (SeD + PTSD), and with PTSD alone, in order to find potential markers of altered stress reactivity associated with both depression and PTSD.

## 2. Materials and Methods

### 2.1. Study Design and Participants

The study involved 460 participants. Out of them, 420 subjects with different levels of depression severity were included in the study group and 40 subjects (20 males and 20 females) in the control group. The mean age was 45.2 ± 4.5 years old (range: 19–47 years). Each study group consisted of 60 patients (30 males and 30 females) with mild depression (MD), moderate depression (MOD), severe depression (SeD), MD and PTSD (MD + PTSD), MOD and PTSD (MOD + PTSD), SeD and PTSD (SeD + PTSD), and with PTSD alone.

The recruitment process took place between 2012 and 2016. The fourth edition of the Diagnostic and Statistical Manual of Mental Disorders (DSM-4) was utilized to diagnose PTSD between 2012 and the first half of 2015. In this period, 253 patients were enrolled in the study group and 20 in the control group. The fifth edition of the Diagnostic and Statistical Manual of Mental Disorders (DSM-5) was adopted between the second half of 2015 and 2016. In this period, 167 patients were enrolled in the study group and 20 in the control group.

According to the DSM-5 criteria, it is possible to diagnose PTSD using four diagnostic clusters such as intrusion, avoidance, negative cognitions and mood, as well as changes in arousal and reactivity [10]. The avoidance of previously experienced traumatic stimuli by PTSD patients leads to negative changes in cognitions and mood. In addition, they are unable to recall the details of these traumatic events or differentiate between emotions. Changes in arousal and reactivity make these patients show aggressive and self-destructive behaviours. All these symptoms are usually observed for at least one month. The DSM-5 was applied to diagnose depression and its severity was evaluated with the psychometric properties of the Beck Depression Inventory (BDI-II). It is comprised of twenty-one questions with four responses to choose from. This allowed the patients to indicate their feelings over the previous month. The respondents could earn between 0 and −3 points for each answer. The minimum score was 0 and the maximum score was 63. The cut-off values were as follows: 0–11 = no depression, 12–19 = mild depression, 20–25 = moderate depression, over 26 = severe depression.

All patients were hospitalized at the Department of Psychiatry between 2012 and 2016. The control group comprised forty sex-age-matched healthy individuals. Their mean age was 42.4 ± 4.1 years old (range: 23–48 years). Patients with the following conditions were excluded from the study: other mental disorders, CNS injury, neurological or neurodegenerative disorders, alcohol or other substance dependence, infectious and chronic somatic comorbidities, smoking, and taking medications. Moreover, women or controls had no menopause. The first author of this paper—a specialist in psychiatry and family medicine—was responsible for evaluation by using the DSM-5 and the Hamilton Depression Rating Scale.

All subjects had fifteen millimetres of blood collected on the first day of their admission before pharmacological therapy. As far as females are concerned, blood was collected during the follicular phase of the menstrual cycle. All subjects had not been on any medication for two weeks before blood drawing to make the test results more reliable.

All PTSD subjects were in the 3rd phase (intrusive/repetitive stage) of the disease. This phase is characterized by nightmares, flashbacks and restlessness despite denial. It is considered the most destructive phase of all PTSD stages. It is important to note that PTSD patients are eager to confront PTSD.

### 2.2. Determination Methods

The subjects had their blood samples collected between 7 and 9 in the morning and sent for laboratory tests. The procedure involved centrifugation at 3500 rpm for 10 min and then placing the samples in 2 mL tubes for storage at −70 °C. The following markers were measured with the ELISA method: IL-4, (Diaclone SAS, Besancon, France); IL-8 (Diaclone SAS, Besancon, France); IL-10, (Diaclone SAS, Besancon, France); IL-1β, (Diaclone SAS, Besancon, France).

The manufacturer’s instructions were used to perform lab analysis. This involved checking the results and measuring concentrations. After plates had been coated with monoclonal antibodies for selected markers, the plasma samples were incubated for 60 min. Some of them bound to specific antibodies and all unbound antibodies were removed. The samples were incubated for 30 min. Afterwards, a colour reaction was induced, depending on the concentration levels.

The lower concentration levels of studied cytokines were IL-4, (0.31 pg/mL); IL-8, (29.0 pg/mL); IL-10, (4.9 pg/mL); IL-1β, (6.5 pg/m). If necessary, technical replicates were used in the study. All samples were assayed at the same time.

### 2.3. Statistical Analysis

The Statistica 10.0 commercial package was used to conduct a statistical analysis. The findings are presented as the standard error of measurement (SEM).

The two-way analysis of variance (ANOVA) was adopted for the study. The Shapiro-Wilk or the Mann-Whitney U-test were adopted to compare the mean values. Statistical significance was considered at *p* < 0.05. The Kruskal–Wallis ANOVA test was used to perform a comparative analysis of three or more independent groups with distribution other than normal.

If the null hypothesis on the equality of the variance of the studied groups was rejected, a post hoc analysis was performed.

### 2.4. Ethics

The study received the approval of the Research Ethics Committee, Faculty of Medicine, Collegium Medicum in Bydgoszcz (KB/193/2012; KB/445/2016). Written informed consent was provided by all subjects before the study. An interview was conducted with all subjects and their mental status was evaluated by a psychiatrist—the author.

## 3. Results

The study of the mean concentration levels of interleukin 1β (IL-1β), using the ANOVA analyses, revealed statistically significant differences between the control group and the study groups. The analysis was performed for the control group [comprising 40 subjects] and for the study groups described in the Material and Methods section [each group comprised 60 subjects]. In the first stage, the Mann-Whitney U-test was performed to check whether the values of samples collected from two independent populations were equally large. The hypotheses were verified at the significance level of = 0.05. Based on the adopted significance level and the statistics of the Mann-Whitney U-test, without correction for continuity (*p* = 0.015), as well as with this correction (*p* = 0.016), and based on the exact Ustatistic (*p* = 0.015), statistically significant differences were found between the studied groups, the size of which is considered representative.

It was reported that the mean concentration levels of IL-1β increased as depression became more severe. When PTSD co-occurred with depression, the concentration levels of IL-1β were observed to increase even more significantly. According to the ANOVA analyses, MD subjects were found to have lower IL-1β concentration levels (1.23 ± 0.21 pg/mL) than MD+PSTD subjects (2.39 ± 0.41 pg/mL) [mean and standard deviation analysed with the significance level of = 0.05 for the MD group is shown in Table 1]. Thus, the coexistence of PTSD elevated IL-1β concentration levels in MD subjects by 94%. For MOD subjects, this increase was by 58%, and for SeD subjects by 13%.

The concentration levels of IL-1β in SeD patients were found to be higher when compared to MD and MOD patients, as well as MD + PTSD and MOD + PTSD patients (mean and standard deviation analysed with a significance level of = 0.05 for the MD + PTSD and MOD + PTSD groups is shown in Table 1).

The only subjects reported to have higher IL-1β concentration were patients with SeD + PTSD.

Interleukin-1β concentration levels were statistically significantly higher than those observed in the control group (marked as group C in Table) (*p* < 0.001 except for the group with PTSD), (Table 1, Figure 1).

Another analysed parameter was interleukin-8 (IL-8), the level of which changed in a way similar to the previous marker. The differences between the mean concentration levels of IL-8 were statistically significant in all study groups when compared to the control group (*p* < 0.001).

Similarly, IL-8 concentration levels also increased as depression intensified. Its cooccurrence with PTSD elevated IL-8 concentration levels, the progression of which depended on depression severity.

While in the case of MD subjects, the concentration of IL-8 was 0.95 ± 0.31 pg/mL, in the group of MD + PTSD subjects, it was 1.40 ± 0.19 pg/mL (47% higher). In the group of MOD patients, the concentration of IL-8 was 1.49 ± 0.19 pg/mL, while in the group of MOD + PTSD patients, it was already 2.31 ± 0.30 pg/mL (53% higher). Similarly, for SeD patients, IL-8 concentration levels were elevated by 94%—from 1.79 ± 0.38 pg/mL in SeD subjects to 3.54 ± 0.26 pg/mL in SeD + PTSD subjects (Table 1, Figure 2).

Similar results were observed when analysing interleukin-4 (IL-4) concentration levels. Again, its concentration increased as depression progressed. Additionaly, the impact of PTSD was crucial, as it significantly increased IL-4 concentration levels. A comparative analysis showed significant differences between individual study groups and the control group (*p* < 0.001). It is worth noting that the concentration of IL-4 in the group of patients with PTSD alone was highest in all compared study groups (13.81 ± 0.44 pg/mL). This situation failed to be reported in any other analysed parameters (Table 1, Figure 3).

It is important to look at the concentration of interleukin-10 (IL-10) in individual study groups. A regular decrease in IL-10 concentration levels was noted with both the onset and exacerbation of depressive symptoms. In the control group, the mean concentration of IL-10 was 45.54 ± 2.11 pg/mL, while in the MD group, it was 40.23 ± 1.90 pg/mL. In the MOD group, the result was 35.88 ± 2.47 pg/mL, and in the SeD group, it was already 29.62 ± 1.91 pg/mL. Therefore, it can be concluded that the concentration dropped regularly by about 5 pg/mL with each stage of depression.

Even more crucial findings could be observed when depression coexisted with PTSD. Particularly unusual was the result found in the MD + PTSD group (80.88 ± 4.07 pg/mL). It was almost twice as high as that observed in the control group (45.54 ± 2.11 pg/mL). In the MOD + PTSD group, the concentration level of IL-10 was 26.19 ± 1.35 pg/mL, and in the case of SeD + PTSD patients, it was 14.49 ± 1.74 pg/mL (Table 1, Figure 4).

When taking the gender of subjects into account, interesting observations were made based on a comparison of individual concentration levels of four blood parameters. As Table 1 shows, when contrasting the significance of the mean concentration levels of four analysed parameters between males and females, different results can be noted.

In the case of IL-1β, significant differences between males and females were found in all patient groups, except those with MOD + PTSD (*p* = NS). In this case, females had higher concentrations of IL-1β than males each time.

For IL-8, the situation was slightly different. Significant differences between males and females were reported only in three groups of patients. They include MD, SeD and MOD + PTSD groups. In each of these groups, females were found to have higher concentration of IL-8 than males. In the remaining study groups, the differences between males and females were not statistically significant (*p* = NS).

As far as IL-4 is concerned, the situation was different than in the previous parameter. Here, significant differences in IL-4 concentration levels between males and females were shown in almost all groups, except for MOD and MD + PTSD subjects. In almost every group, females were observed to have elevated concentration levels of IL-4 versus males. The exception was the group of patients with PTSD alone. In this case, males were reported to have higher concentration levels than females.

For IL-10, significant differences were observed between males and females in all study groups except those with MD and PTSD. Females from the control group and with MD were found to have elevated levels of IL-10 versus males. In turn, in the remaining groups (MOD, SeD, MOD + PTSD and SeD + PTSD), males were reported to have higher IL-10 concentration levels than females (Table 1).

Based on the multi-factor analysis of variance (ANOVA) that looked at depression, PTSD and gender, the above findings were confirmed. The null hypothesis was rejected at *p* < 0.001 and it claimed that the studied factors had an impact on either depression or PTSD.

Addictionaly, all levels of depression severity and PTSD, with and without depression, were found to have an influence on the studied blood parameters (*p* < 0.001).

The null hypothesis claiming that gender had no impact on the disorders was also rejected at *p* < 0.001 in the case of three parameters, such as interleukine 1β (IL-1β), interleukine-8 (IL-8) and interleukine-4 (IL-4). For interleukin-10 (IL-10), there is no rationale to reject the null hypothesis, which means no effect in terms of gender.

## 4. Discussion

The present study looked at the serum concentration of IL-4, IL-8, IL-10, and IL-1β in both males and females suffering from different types of depressive episodes (mild, moderate, and severe), with and without comorbid PTSD. This study is a continuation of the author's previous research on the involvement of cytokines in depressive disorders [12,13,14]. The authors of this study found significant differences in the concentration of IL-1, IL-4, and IL-10 between males and females in the control group. The results suggest the existence of sexual dimorphism in the context of the immune response. This indicates the existence of a relationship between gender and certain gene polymorphisms for proinflammatory cytokines. Songtachaler T. et al. and Goel N. et al. point to the existence of a gender-specific pattern related to the immune response [20,21].

The study of the mean concentration levels of IL-1β showed the presence of statistically significant differences (*p* < 0.001) between the control group and other study groups. The concentration of IL-1β increased with the severity of depression and when concurrent with PTSD. Additonally, it was higher in SED + PTSD subjects when compared to other groups—both with and without coexisting PTSD. When analysing the concentration of IL-1β, it was shown that females had higher concentration levels when compared to males.

Zhang HX. et al. found an increase in IL-1β among SeD patients [22]. Another study by Al-Hakeim HK. et al. reported a significant increase (*p* < 0.05) in the serum concentration of IL-1β, IL-4 and insulin resistance (HOMA2IR) in 63 patients with major depressive disorder (MDD) when compared to the controls [23]. Farooq RK. et al., also studied IL-1β and found that it increased in depressed elderly subjects, which was directly proportional to the severity of illness [24]. In another study, IL-1β concentration levels were observed to be elevated in females with depressive symptoms versus those who did not have any such reactions 1-month post-partum [25]. Other research, for example by Wang W. et al. or Wieck A. et al., concerned cytokines in PTSD [15,26]. Wang W. et al. (2019) [15] showed an increase in the level of IL-1β in PTSD patients exposed to a deadly earthquake. Wieck et al. studied neuroimmunoendocrine interactions in PTSD with special focus on the long-term implications of childhood maltreatment [26]. The study assessed the impact of early-life stress (ELS) in adult immunological changes commonly observed in PTSD. Both studies confirm that MDD and PTSD are accompanied by the activation of the immune system with significant elevations in cytokine levels. Additionally, the results indicate the stimulation of the immune system by increased cytokine levels in both MDD and PTSD. Neither study, however, assessed the coexistence of depressive disorders, which was the subject covered by the author of this article.

Additionally, in our study, the analysis of IL-8 concentration levels showed that this parameter was subject to similar changes as in the case of IL-1β. Similarly, the differences between the mean IL-8 concentration levels were statistically significant in all study groups when compared to the control group (*p* < 0.001). Along with the severity of depression, IL-8 concentration levels increased, and the co-occurrence of PTSD also elevated the IL-8 concentration. The largest increase in IL-8 concentration levels (by as much as 94%) was found in the SeD + PTSD group. However, a decrease in the level of IL-8 was observed in the MOD + PTSD and SED + PTSD groups versus the PTSD group. This result indicates the significant impact of the severity of depression on IL-8 concentration levels. Significant differences between males and females were found in IL-8, yet only in three groups of patients: MD, SeD, and MOD + PTSD. In each of these groups, females were reported to have higher concentration levels of IL-8 when compared to men. In the remaining study groups, the differences between males and females were not statistically significant (*p* = NS).

Vogelzangs N. et al. showed an increase in IL-8 concentration levels in patients with anxiety-depressive disorder [16]. In addition, they found a relationship between IL-8 and somatic and cognitive anxiety symptoms. In addition, Baune BT. et al. found elevated serum levels of IL-8 in the group of older patients with MD and MOD [27].

Chen CY. et al. studied 91 MDD patients and reported a decrease in IL-8 and IL-1β concentration levels during venlafaxine and paroxetine treatment after 8-weeks as well as a decrease in the number of points in the Hamilton Depression Rating Scale [28]. It was also revealed that venlafaxine treatment caused greater decreases than paroxetine. Since inflammatory processes play a crucial role in the pathophysiology of depression, it is important to identify specific cytokines targeted by different antidepressants for personalized treatment.

Similarly, to IL-1β and IL-8, the analysis of IL-4 demonstrated that its concentration levels increased as depression became more severe, regardless of whether with or without concomitant PTSD. The comparative analysis of the mean values showed significant differences in IL-4 concentration levels in individual groups versus the control group (*p* < 0.001). The highest IL-4 concentration level was found in the group with PTSD. All study groups in which PTSD coexisted with depression showed lower IL-4 concentration levels than those with PTSD alone. This may show that in the case of IL-4, PTSD alone has a greater impact on increased serum levels of this cytokine compared to patients with depression concomitant with PTSD. Additionally, significant differences in IL-4 concentration levels were reported between males and females in all groups except for MOD and MD + PTSD patients. In almost every group, females were observed to have higher IL-4 concentration levels than men. The exception was the PTSD group in which males were found to have increased IL-4 concentration levels when compared to females.

The study by Twayej AJ. et al. showed the upregulation of IL-4, pointing to the activation of the immune-inflammatory response system (IRS) and the compensatory immune regulatory system (CIRS) [29]. These results indicate that the immune system becomes stimulated as well as IR increased and modulated due to elevated cytokine concentration levels observed in MDD. Similarly, Al-Hakeim HK. et al. assessed IL-4 and IL-1β in 63 MDD patients and 27 healthy controls [23]. As a result, a significant increase (*p* < 0.05) in the serum levels of IL-1β and IL-4 was found in MDD patients versus the controls. Their study further confirms that MDD is accompanied by activation of the immune system with significant elevations in the concentration of both IL-4 and IL-10. The study also revealed that ECT treatment leads to alterations in cytokine levels. Immune mediators, found in MDD patients, have been reported to be different from those of healthy controls. The decrease in this difference, due to treatment, shows that immune mediators might be useful markers. Guo M. et al. found elevated concentration levels of IL-4 in PTSD patients versus the controls [30]. Similar results were obtained in the research conducted among war veterans with PTSD [31].

By contrast, IL-10 concentration levels regularly decreased as depression progressed. The lowest IL-10 concentration level was reported in the SeD group. It can be stated that the concentration dropped regularly by about 5 pg/mL with each stage of depression. In the MD + PTSD group, however, the IL-10 concentration levels were almost twice as high as in the control group. The high concentration of IL-10 found in MD + PTSD patients in this study would suggest that it acts by suppressing an inflammatory response. In the MOD + PTSD group, the IL-10 concentration level amounted to 26.19 ± 1.35 pg/mL, and in the SeD + PTSD group, it reached the lowest values, amounting to 14.49 ± 1.74 pg/mL.

In addition, significant differences were observed between males and females in all study groups,—except MD and PTSD patients. Females were reported to have higher levels of IL-10 in the control group and in the MD group. In turn, males were observed to have elevated levels of IL-10 in the MOD + PTSD and SeD + PTSD groups.

Taraz et al., found that patients with severe depression treated with haemodialysis had decreased concentration levels of IL-10, which became significantly elevated after 12 weeks of sertraline treatment [32]. The results showed that the serum levels of IL-10 were significantly (*p* = 0.011) decreased in depressed patients (2.8 ± 0.41 pg/mL) versus healthy controls (4.3 ± 0.4 pg/mL). In addition, the decreased IL-10 concentration levels in depressed patients may be responsible for the induction of inflammation [17]. This fact is in harmony with the previous studies which claimed that depressed patients suffer from chronic inflammation.

In a study by Dhabhar FS. et al., the serum concentration of IL-10 was decreased while IL-6 did not change in depressed patients when compared to healthy controls [18]. Li Y. et al. reported fewer T regulatory lymphocytes (the main source of IL-10) in depressed patients versus healthy controls [33]. Song and colleagues revealed that the serum concentration levels of IL-1β were increased and, inversely, IL-10 concentration levels were decreased in depressed patients [19]. Other authors [34,35,36,37] also confirmed these observations.

## 5. Conclusions

The above findings show that long-term stress, such as depression and PTSD, cause the up regulation of inflammatory markers. Chronic stress may elevate stress perception, which induces the production of pro-inflammatory messenger molecules responsible for depression. This knowledge may be useful for future research on anti-inflammatory markers.

## Figures and Tables

**Figure 1 brainsci-12-00387-f001:**
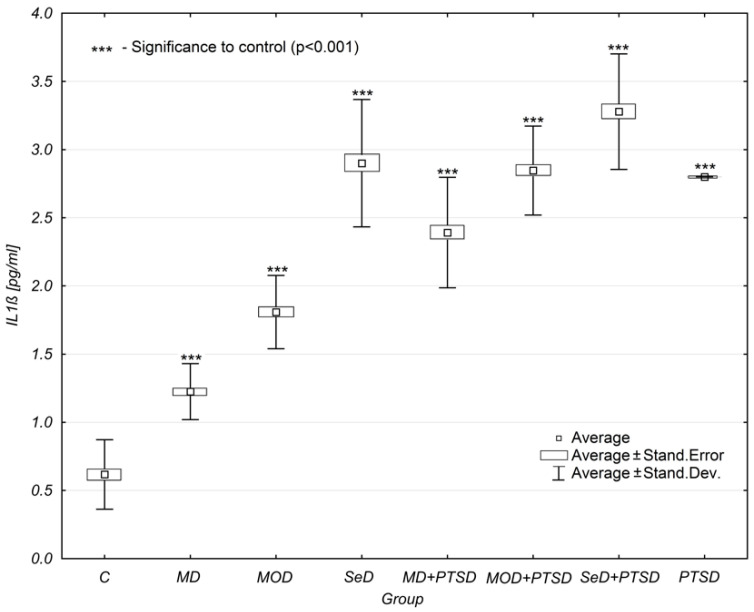
Mean IL-1β concentration levels in selected study groups. Legends: C—control group, MD—mild depression; MOD—moderate depression; SD—severe depression; MD + PTSD—mild depression + posttraumatic stress disorder; MOD + PTSD—moderate depression+posttraumatic stress disorder; SeD + PTSD—severe depression + posttraumatic stress disorder; PTSD—posttraumatic stress disorder; IL-1β—interleukin-1β. Number of patients in study groups (females + males): C = 80, MD = 120, MOD = 120, SD = 120, MD + PTSD = 120; MOD + PTSD = 120; SeD + PTSD = 60; PTSD = 120. *** *p* < 0.005 when compared to the control group. Since all data was distributed normally, the one-way ANOVA test was used for statistical analysis (data is presented in the form of a box plot, which can be used to estimate differences in the mean values; the graphs show the mean standard error).

**Figure 2 brainsci-12-00387-f002:**
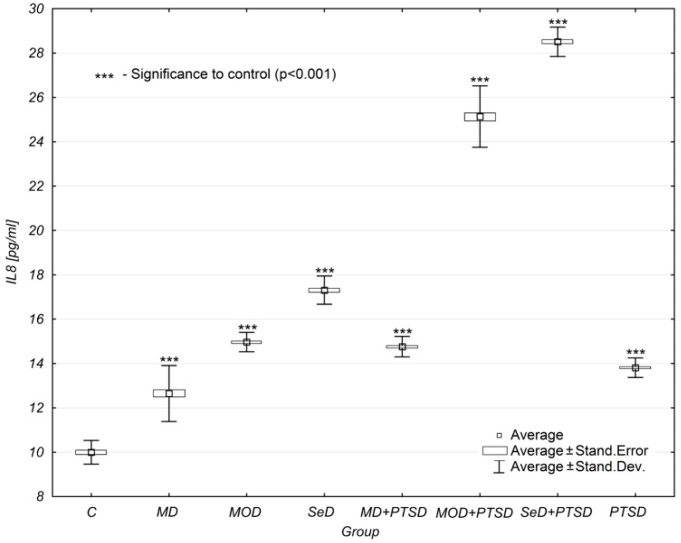
Mean IL-8 concentration levels in selected study groups. Legends: C—control group, MD—mild depression; MOD—moderate depression; SD—severe depression; MD + PTSD—mild depression + posttraumatic stress disorder; MOD + PTSD– moderate depression + posttraumatic stress disorder; SeD + PTSD—severe depression + posttraumatic stress disorder; PTSD—posttraumatic stress disorder; IL-8—interleukin-8. Number of patients in the study groups (females + males): C = 80, MD = 120, MOD = 120, SD = 120, MD + PTSD = 120; MOD + PTSD = 120; SeD + PTSD = 60; PTSD = 120. *** *p* < 0.005 when compared to the control group. Since all data was distributed normally, the one-way ANOVA test was used for statistical analysis (data is presented in the form of a box plot, which can be used to estimate differences in the mean values; the graphs show the mean and standard error).

**Figure 3 brainsci-12-00387-f003:**
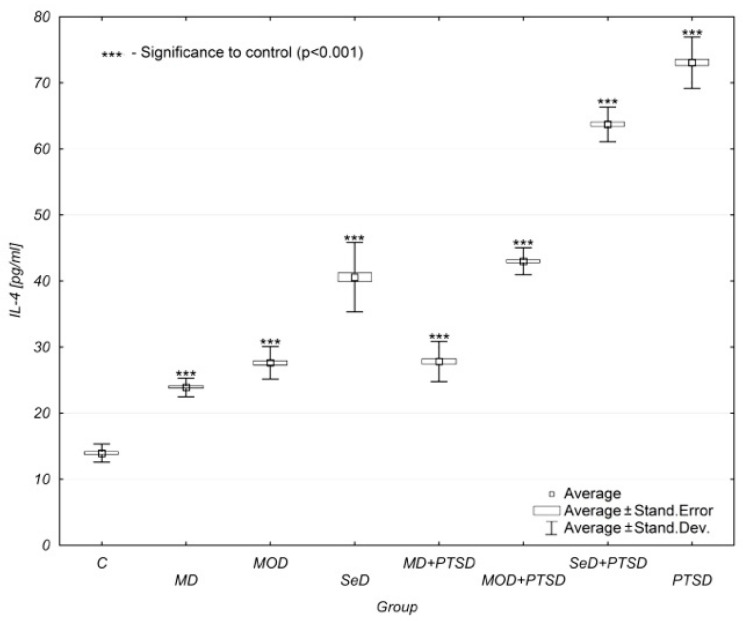
Mean IL-4 concentration levels in selected study groups. Legends: C—control group, MD—mild depression; MOD—moderate depression; SD—severe depression; MD + PTSD—mild depression n+ posttraumatic stress disorder; MOD + PTSD—moderate depression + posttraumatic stress disorder; SeD + PTSD—severe depression+ posttraumatic stress disorder; PTSD—posttraumatic stress disorder; IL-4—interleukin-4. Number of patients in the study groups (females + males): C = 80, MD = 120, MOD = 120, SD = 120, MD + PTSD = 120; MOD + PTSD = 120; SeD + PTSD = 60; PTSD = 120. *** *p* < 0.005 when compared to the control group. Since all data was distributed normally, the one-way ANOVA test was used (data is presented in the form of a box plot, which can be used to estimate differences in the mean values; the graphs show the mean and standard error).

**Figure 4 brainsci-12-00387-f004:**
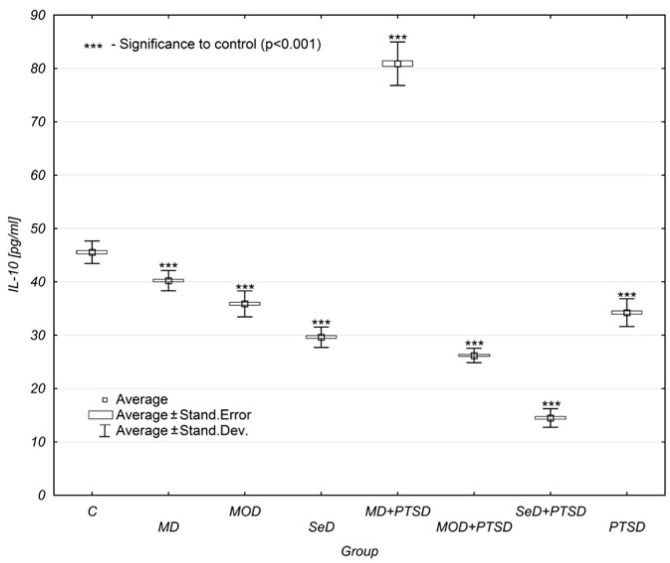
Mean IL-10 concentration levels in selected study groups. Legends: C—control group, MD—mild depression; MOD—moderate depression; SD—severe depression; MD + PTSD—mild depression + posttraumatic stress disorder; MOD + PTSD—moderate depression + posttraumatic stress disorder; SeD + PTSD—severe depression + posttraumatic stress disorder; PTSD—posttraumatic stress disorder; IL-10—interleukin-10. Number of patients in the study groups (females + males): C = 80, MD = 120, MOD = 120, SD = 120, MD + PTSD = 120; MOD + PTSD = 120; SeD + PTSD = 60; PTSD = 120. *** *p* < 0.005 when compared to the control group. Since all data was distributed normally, the one-way ANOVA test was used for statistical analysis (data is presented in the form of a box plot, which can be used to estimate differences in the mean values; the graphs show the mean and standard error).

**Table 1 brainsci-12-00387-t001:** Comparison of IL-1β, IL-8, IL-4, IL-10 between males and female.

Group	Sex	IL-1β [pg/mL]	*p*	IL-8 [pg/mL]	*p*	IL-4 [pg/mL]	*p*	IL-10 [pg/mL]	*p*
C [*n* = 40]	M	0.48 ± 0.17	0.001	0.54 ± 0.2	0.989	9.71 ± 0.5	0.001	44.44 ± 1.93	0.001
	F	0.76 ± 0.25		0.54 ± 0.23		10.28 ± 0.41		46.66 ± 1.7	
MD [*n* = 60]	M	1.16 ± 0.23	0.001	0.70 ± 0.16	0.001	11.65 ± 0.95	0.001	40.57 ± 1.47	0.124
	F	1.29 ± 0.16		1.20 ± 0.2		13.64 ± 0.53		39.89 ± 2.22	
MOD [*n* = 60]	M	1.71 ± 0.21	0.001	1.45 ± 0.15	0.073	14.99 ± 0.38	0.678	37.51 ± 2.15	0.001
	F	1.91 ± 0.29		1.54 ± 0.21		14.95 ± 0.49		34.25 ± 1.43	
SeD [*n* = 60]	M	2.66 ± 0.33	0.001	1.56 ± 0.27	0.001	16.85 ± 0.38	0.001	30.31 ± 1.76	0.005
	F	3.14 ± 0.46		2.03 ± 0.32		17.76 ± 0.51		28.94 ± 1.83	
MD + PTSD [*n* = 60]	M	2.2 ± 0.26	0.001	1.39 ± 0.12	0.922	14.67 ± 0.3	0.494	78.95 ± 4.5	0.001
	F	2.59 ± 0.43		1.4 ± 0.24		14.86 ± 0.56		82.82 ± 2.41	
MOD + PTSD [*n* = 60]	M	2.85 ± 0.40	0.127	2.12 ± 0.25	0.001	23.99 ± 0.98	0.001	26.6 ± 1.3	0.031
	F	2.84 ± 0.23		2.50 ± 0.22		26.28 ± 0.51		25.77 ± 1.3	
SeD + PTSD [*n* = 60]	M	3.1 ± 0.42	0.001	3.56 ± 0.23	0.511	28.19 ± 0.57	0.001	15.32 ± 1.7	0.001
	F	3.45 ± 0.35		3.51 ± 0.28		28.83 ± 0.6		13.64 ± 1.33	
PTSD [*n* = 60]	M	2.80 ± 0.00	-	1.46 ± 0.16	0.326	14.04 ± 0.38	0.001	33.88 ± 2.82	0.126
	F	2.80 ± 0.00		1.52 ± 0.21		13.57 ± 0.36		34.59 ± 2.34	

Legends: M—men; W—women; *p*—significance level *p* 0.05; C—control group, MD—mild depression; MOD—moderate depression; SD—severe depression; MD + PTSD—mild depression + posttraumatic stress disorder; MOD + PTSD—moderate depression + posttraumatic stress disorder; SeD + PTSD—severe depression + posttraumatic stress disorder; PTSD—posttraumatic stress disorder; IL-1β—interleukin-1β, IL-8—interleukin-8; IL-4—interleukin- 4; IL-10—interleukin-10. Since all data was not distributed normally, the Kruskal–Wallis test was used for statistical analysis. χ2 = 14.07, H = 1.82, for ANOVA Fkr = 3.79.

## Data Availability

The datasets used and analyzed during the current study are available from the corresponding author on reasonable request.

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
