# Peer review of "Changes in the Serum Levels of Cytokines: IL-1β, IL-4, IL-8 and IL-10 in Depression with and without Posttraumatic Stress Disorder"

_brainsci, 2022, doi:10.3390/brainsci12030387_

Round 1

Reviewer 1 Report

Line 73 It is noted that the pro-inflammatory IL-6 was not included among the cytokines tested. Rationale as to why IL-6 was overlooked or excluded from the list of cytokines should be mentioned. The only place IL-6 is mentioned in the manuscript is on line 478 where IL-6 was found not to change in depressed individuals. If you analyzed IL-6 and the findings were not significant that is important to include. In our experience we have had issues with IL-6 in animal serum, so that is why the lack of IL-6 inclusion in this study caught my attention.

Line 121 “As many as 460” should be rephrased, the number of enrolled participants should be a definitive number. If it was 460 then say 460 or however many were enrolled.

Lines 122-123 Include the word “the mean age of “all “study participants was 45.2 yo (range 19-47) if that is the case. This age and range follows a sentence talking about the control group consisting of 20 males and 20 females, so it makes logical sense to include the age of the control group (and range of ages), here, however lines 153 states that the control group mean age was 42.4 (range 23-48).  So line 123 needs to have the word “all” or “overall” if pertaining to all 420 participants.  Please clarify.

Line 177 were the specific cytokines tested part of a larger ELISA kit (many only IL-1B, IL-4, IL-8, IL-10 were selected to be analyzed for this study. A sentence would be beneficial describing other cytokines in the ELISA kit that were not evaluated in this study.

Line 234- Compared to what (the control group?)

Line 263 Clarify what is “previous marker” as two were discussed prior to this sentence. I assume it was IL-1B but state specifically.

Fig. 2 It is interesting to note that PTSD alone had much lower IL-8 compared to the other cytokines, this difference might be highlighted better in the discussion, as it seems contrary to the other cytokines.

Line 346- Clarify “previous parameters” specifically

Lines 372-373 Need to capitalize “IL” at several locations

Line 376 It is an ackward beginning to a sentence. Perhaps begin with “The existence of a gender-specific pattern related to the immune response has been demonstrated (refs).

Author Response

Dear Editor,

In response to the review of the paper titled: ‘Changes in the serum levels of cytokines: IL-1β, IL-4, IL-8 and IL-10 in Depression with and without Posttraumatic Stress Disorder’ (Manuscript ID: brainsci-1527805), the relevant changes suggested by the editor have been made.

In accordance with the recommendations of Reviewer 1:

    1. Line 73: The reference to IL-6 has been removed from the text. IL-6 was not studied in this article. The author has studied IL-6 in patients with PTSD in the following publications:
  1. Ogłodek EA, Szota AM, Moś DM, Araszkiewicz A, Szromek AR. Serum concentrations of chemokines (CCL-5 and CXCL-12), chemokine receptors (CCR-5 and CXCR-4), and IL-6 in patients with posttraumatic stress disorder and avoidant personality disorder. Pharmacol Rep. 2015 Dec;67(6):1251-8. doi: 10.1016/j.pharep.2015.05.023. 
  2. Ogłodek EA, Szota A, Just MJ, Moś D, Araszkiewicz A. Comparison of chemokines (CCL-5 and SDF-1), chemokine receptors (CCR-5 and CXCR-4) and IL-6 levels in patients with different severities of depression. Pharmacol Rep. 2014 Oct;66(5):920-6. doi: 10.1016/j.pharep.2014.06.001.
    1. Line 121: The sentence: 'As many as 460 participants were enrolled in the study” has been changed into: ‘A total number of participants enrolled in the study was 460’

Lines 122-123: The sentence: ‘The mean age of the study group was 45.2±4.5 years old (range: 19-47 years)’ has been changed into: ’The mean age of all participants was 45.2±4.5 years old (range: 19-47 years)’.

Line 153: the sentence: The mean age of the control group was 42.4 ± 4.1 years old (range: 23–48 years) has been deleted.

    1. Line 177: In the determination methods section, text has been added: ‘In her previous studies, the author determined other cytokines by ELISA in psychiatric disorders with a depressive-anxiety spectrum: chemokines CCL-5 and CXCL-12, chemokine receptors (CCR-5 and CXCR-4), and IL-6 (PTSD + personality disorder - PeD);  CCL-5/RANTES), CXCL-12/SDF-1 (PD+PeD); iNOS, HO-1, IL-33, MIP-1β (DD+ PTSD);  PON-1, GR, IL-18, and OxLDL (DD+PTSD); NT-4/5, GPX-1, TNF-α, and l-arginine (DD+PTSD); chemokines (CCL-5 and SDF-1), chemokine receptors (CCR-5 and CXCR-4) and IL-6 (DD); neurotrophins NT-3, BDNF, NGF (DD); chemokine receptors (CXCR-5 , CXCR-4 -and IL-6 (DD); chemokines (MCP-1, CCL-5 and SDF-1) (generalized anxiety disorder – GAD+PeD). References to the author's publications have been provided.
    2. Line 234: The sentence: ‘The only subjects who were reported to have higher IL-1β concentration levels were patients with SeD+PSTD’ has been changed into: ‘The concentration levels of IL-1β in SeD+PTSD patients were found to be higher when compared to SeD patients, as well as MD+PTSD and MOD+PTSD patients.’
    3. Line 263: The sentence: ‘Another analyzed parameter was interleukin-8 (IL-8), the level of which changed in a way similar to the previous marker’ has been changed into: ‘Another analyzed parameter was interleukin-8 (IL-8), the level of which changed in a way similar to the IL-1β’.
    4. Line 426-427: A sentence has been added in reference to reviewer's comment regarding discussion of Fig. 2: ‘It is interesting to note that PTSD alone had much lower IL-8 when compared to the other cytokines determined in this is study, namely IL-4, IL-10, and IL-1β’.
    5. Line 348: The sentence: ‘As far as IL-4 is concerned, the situation was different than in the previous parameter’ has been changed into: ‘As far as IL-4 is concerned, the situation was different than in the IL-8’.
    6. Lines 372-373: “IL” at several locations has been changed.
    7. Lines 176-185: Previous cytokine studies performed by the author of this article has been described. The literature has been supplemented.
    8. Line 376: the sentence “…The results suggest the existence of sexual dimorphism in the context of the immune response” has been changed into: “The authors of this study found significant differences in the concentration of IL-1, IL-4, and IL-10 between males and females in the control group, which suggests sexual dimorphism in the context of the immune response.

Reviewer 2 Report

21 December 2021

Manuscript ID: brainsci-1527805

Type: Article

Title: “Changes in the serum levels of cytokines: IL-1β, IL-4, IL-8 and IL-10 in Depression with and without Posttraumatic Stress Disorder” by OGŁODEK E., submitted to Brain Sciences

Dear Authors,

Major depressive disorders (MDD) and posttraumatic stress disorder (PTSD) have been linked to inflammation and particularly proinflammatory cytokines. The authors studied interleukin (IL)-1β, IL-4, IL-8 and IL-10 concentrations in the serum of patients with various severity of MDD and PTSD comorbidity by ELISA. The results showed that the levels of IL-1β, IL-4, and IL-8 were elevated, while the levels of IL-10 were lower all groups except for MDD-PTSD comorbidity, compared to healthy controls. The authors concluded that inflammatory factors are potential targets and treatment of MDD and PTSD.

Please reconsider the following:

  1. A graphical abstract summarizing the manuscript is highly recommended.
  2. Page 1, Abstract:
    1. Please present a rationale using the abbreviation DD and choosing the cytokines.
    2. Inflammation is linked to not only MDD and PTSD, but also to many other psychiatric disorders.
    3. “reflected”, “noted”: Please rephrase them to make the results clear.
    4. The cytokines studied are not particularly associated with chronic inflammation. Please rephrase the conclusion.
  3. Page 1, Keywords: Please list up to ten keywords.
  4. Page 2, Introduction:
    1. Paragraph 1: Please briefly describe the epidemiology of MDD and PTSD.
    2. “the HPA axis”: Please define the abbreviation in the first appearance. The link between the hypothalamic–pituitary–adrenal axis and the tryptophan-kynurenine metabolic system is discussed recently.
    3. Paragraph 2: Inflammation is linked not only to MDD and PTSD, but also to many other psychiatric disorders. The link between inflammatory cytokines and major psychiatric disorders is discussed recently. Kynurenines are potential prognostic and therapeutic biomarkers for MDD.
    4. IL-1β and IL-8 (proinflammatory), IL-4 and IL-10 (pleiotropic anti-inflammatory cytokines): Please clearly present proinflammatory or anti-inflammatory cytokines and a rationale to choose those cytokines for this study.
    5. Please rearrange the contents of introduction and detailed discussion can be placed in Discussion.
    6. Suggested refences: doi: 10.17219/acem/139572; doi: 10.3390/biomedicines9070734; doi: 10.1038/s41598-020-73918-
  5. Pages 5-12, Results: Please present figures in color.
  6. Pages 10-12, Discussion: The background is well discussed and compared to this study. However, it deserves to discuss the limitation, weakness, potentials in the study, the ultimate goal, research or knowledge needed to achieve, the future research direction, and the biggest challenge in this goal, among others.
  7. References: Please cite more references, preferably more than 50 for original articles. 

The manuscript contains four figures, one table and 37 references. The manuscript carries important value presenting cytokines as potential therapeutic biomarkers for MDD and PTSD. Thus, I would recommend this manuscript for publication after major revision.

Best regards

Author Response

Dear Editor,

In response to the review of the paper titled: ‘Changes in the serum levels of cytokines: IL-1β, IL-4, IL-8 and IL-10 in Depression with and without Posttraumatic Stress Disorder’ (Manuscript ID: brainsci-1527805), the relevant changes suggested by the editor have been made.

In accordance with the recommendations of Reviewer 2:

  1. A graphical abstract summarizing the manuscript has been done as recommended by the reviewer.
  2. Page 1 Abstract:
  1. The author of this article used the abbreviation (DD) for depressive disorders, both in the abstract and throughout the text.
  2. The sentence: 'Both depressive disorders (DD) and posttraumatic stress disorder (PTSD) are caused by immune system dysfunction’ has been changed into: ‘Many psychiatric disorders, such as depression and post-traumatic stress disorder (PTSD), can be caused by the immune system dysfunction’.
  3. The sentence with words ‘reflected’, ‘noted’ in the ’results’ section has been modified.
  4. No such sentence as: ‘The cytokines studied are not particularly associated with chronic inflammation’ has been found in the paper, as suggested by the Reviewer.
  1. Page 1: 5 Keywords has been added
  2. Page 2:  Introduction:
  1. Paragraph 1: Epidemiology of MDD and PTSD has been briefly described: ‘Overstimulation of IDO leads to the depletion of plasma concentrations of tryptophan (TRP) and, therefore, to the reduced synthesis of 5-HT in the brain, which may play a role in the development of depressive symptoms [2,3]’.
  2. “The HPA axis” abbreviation has been defined.
  3. Paragraph 2: the sentence: ‘Kynurenine causes both anxiety and depression’ has been changed into: ‘Kynurenine causes many psychiatric disorders such as anxiety disorders and depression’.
  4. At the end of ‘Introduction’, the following has been added: ‘In this study, the authors determined cytokines described in the literature as: pro-inflammatory - IL-1β; proinflammatory or anti-inflammatory: IL-8 and IL-4; and anti-inflammatory - IL-10. The choice of these diagnostic parameters was due to the previous studies of these parameters, which were measured in subjects suffering from depression comorbid with PTSD.
  5. The author decided not to change anything more in the 'Introduction' section, which also remains compatible with the comments of the second reviewer.
  • The author has placed the suggested article in the literature.
  • Pages 5-12, Results: The author wants to leave figures in black and white.
  • Pages 10-12, Discussion: Limitations have been added:The present study includes important limitations mainly due to the small number of determined cytokines. Future research should focus on other proinflammatory and anti-inflammatory cytokines. Moreover, future research should also consider individuals with PTSD and depression treated with different antidepressants. This new spectrum of research will contribute to the knowledge of the pathogenesis of both depression and PTSD’.
  • References have been added and renumbered according to the reviewer's comments.

Round 2

Reviewer 2 Report

6 February 2022

Manuscript ID: brainsci-1527805

Type: Article

Title: “Changes in the serum levels of cytokines: IL-1β, IL-4, IL-8 and IL-10 in Depression with and without Posttraumatic Stress Disorder” by OGŁODEK E., submitted to Brain Sciences

Dear Authors,

Major depressive disorders (MDD) and posttraumatic stress disorder (PTSD) have been linked to inflammation and particularly proinflammatory cytokines. The authors studied interleukin (IL)-1β, IL-4, IL-8 and IL-10 concentrations in the serum of patients with various severity of MDD and PTSD comorbidity by ELISA. The results showed that the levels of IL-1β, IL-4, and IL-8 were elevated, while the levels of IL-10 were lower all groups except for MDD-PTSD comorbidity, compared to healthy controls. The authors concluded that inflammatory factors are potential targets and treatment of MDD and PTSD. The author addressed her response, but the manuscript is partially revised. Please reconsider the following:

  1. Page 1, Abstract:
    1. Please present a rationale choosing the cytokines. The author did not address her response to this point and no revision is made.
    2. The cytokines studied are not particularly associated with chronic inflammation. This a Reviewer’s comment. There is no background presented regarding the cytokines studied and relationships between the cytokines and chronic inflammation. So, the author needs to present a solid conclusion based on the background and results. Please rephrase the conclusion.
  2. Page 1, Keywords: Please list up to ten keywords. Suggested keywords: inflammation; biomarker.
  3. Page 2, Introduction:
    1. Paragraph 1: Please briefly describe the epidemiology of MDD and PTSD. The author presented the pathogenesis but did not present the epidemiology of MDD and PTSD.
    2. “(Han et al. 2015)”: Please follow the reference style.
    3. Pageragraph2: “TRYCAT”; Please expand the abbreviation for the first appearance.
    4. Immune cytokines like IL-1β, IL-4, IL-8 and IL-10”; Please present references.
    5. “IL-1β, IL-4, IL-8 and IL-10”; Please expand the abbreviation for the first appearance and use only the abbreviation afterwards.
    6. Regarding comorbidity, kynurenines as biomarker, suggested refences are: https://doi.org/10.3390/biomedicines9050517 doi: 10.1038/s41598-020-73918-z; doi: 10.1016/j.neubiorev.2020.08.010
  4. Pages 5-12, Results: Please present a rationale to present figures in black and white.
  5. Pages 10-12, Discussion: The revision of this section remains minimal. Please expand the discussion presenting the author’s expert view regarding the limitation, weakness, potentials in the study, the ultimate goal, research or knowledge needed to achieve, the future research direction, and the biggest challenge in this goal, among others.
  6. References: Please cite more references, preferably more than 50 for original articles. Suggested references: https://doi.org/10.3390/bs11080110.

The manuscript contains four figures, one table and 48 references. The manuscript carries important value presenting cytokines as potential therapeutic biomarkers for MDD and PTSD. Thus, I would recommend this manuscript for major revision.

Author Response

(The authors gave the same response as above.)
